# Occurrence of *Chlamydiaceae* and *Chlamydia felis pmp9* Typing in Conjunctival and Rectal Samples of Swiss Stray and Pet Cats

**DOI:** 10.3390/pathogens10080951

**Published:** 2021-07-28

**Authors:** Michelle Bressan, Antonella Rampazzo, Jasmin Kuratli, Hanna Marti, Theresa Pesch, Nicole Borel

**Affiliations:** 1Institute of Veterinary Pathology, Vetsuisse Faculty, University of Zurich, 8057 Zurich, Switzerland; michelle.bressan@uzh.ch (M.B.); jasmin.kuratli@uzh.ch (J.K.); theresa.pesch@uzh.ch (T.P.); n.borel@access.uzh.ch (N.B.); 2Center for Clinical Studies, Vetsuisse Faculty, University of Zurich, 8057 Zurich, Switzerland; 3Ophthalmology Unit, Vetsuisse Faculty, University of Zurich, 8057 Zurich, Switzerland; antonella.rampazzo@uzh.ch

**Keywords:** *Chlamydiaceae*, *Chlamydia felis*, stray cat, pet cat, conjunctivitis, polymorphic membrane protein, *pmp9*, rectal shedding, single-nucleotide polymorphism

## Abstract

*Chlamydia* (*C.*) *felis* primarily replicates in feline conjunctival epithelial cells and is an important cause of conjunctivitis in cats. Data on *C. felis* infection rates in stray cats in Switzerland has been missing so far. We performed a qPCR-based *Chlamydiaceae*-screening on 565 conjunctival and 387 rectal samples from 309 stray and 86 pet cats followed by *Chlamydia* species identification and *C. felis* typing using the gene *pmp9*, which encodes a polymorphic membrane protein. Overall, 19.1% of the stray and 11.6% of the pet cats were *Chlamydiaceae*-positive with significantly higher rates in cats displaying signs of conjunctivitis (37.1%) compared to healthy animals (6.9%). Rectal shedding of *Chlamydiaceae* occurred in 25.0% of infected cats and was mostly associated with concurrent ocular positivity (87.5%). In 92.2% of positive conjunctival and rectal samples, the *Chlamydia* species was identified as *C. felis* and in 2.6% as *C. abortus*. The *C. felis pmp9* gene was very conserved in the sampled population with only one single-nucleotide polymorphism (SNP) in one conjunctival sample. In conclusion, *C. felis* strains are circulating in Swiss cats, are associated with conjunctivitis, have a low *pmp9* genetic variability, and are rectally shed in about 16% of positive cases.

## 1. Introduction

The *Chlamydiaceae* family consists of Gram-negative obligate intracellular bacteria and includes a single genus *Chlamydia*, which comprises 14 species to date, causing a wide range of diseases in different host species including humans, companion animals, livestock, wildlife, and exotic animals [1,2]. *Chlamydia* (*C.*) *felis* is an important cause of conjunctivitis in cats [3]. It is primarily a pathogen of the conjunctiva and possibly the nasal epithelium [4,5]. Furthermore, it has also been detected in the reproductive tract as well as in different organs and tissues such as the lungs, tonsils, gastric mucosa, spleen, liver, kidneys, and peritoneum (reviewed in [4,6,7,8]). The biological significance of these latter findings, however, is unclear.

The *C. felis* genome is highly conserved, which has limited the application of classical typing methods (described in Section 3.4.) and implicated the examination of other typing targets. Polymorphic membrane proteins (Pmps) are a family of surface proteins that are suspected to be involved in different processes of chlamydial infection such as tissue tropism, adhesion, modulation of host immune responses, and pathogenesis [9,10,11,12]. Given their high degree of inter- and intra-species variation [13], they are promising candidates for typing of highly conserved chlamydial species with low rates of recombination. Moreover, some degree of genetic variation has already been described for the *C. felis pmp9* gene, which we aimed to investigate in this study [14].

*C. felis* is the main *Chlamydia* species associated with cats, although other chlamydial species, in particular *C. abortus* [15], *C. pneumoniae* [16], *C. psittaci* [17,18], and *C. suis* [18] have occasionally been reported as well. Moreover, *Chlamydia*-like organisms of the *Parachlamydiaceae*-family, especially *Neochlamydia hartmannellae* and *Parachlamydia acanthamoebae*, have been detected in cats with and without ocular disease [4,13,14].

*C. felis* features a relatively high host specificity compared to other members of the *Chlamydiaceae* family. Nevertheless, it has also been detected in dogs and was recently found in a free-ranging Eurasian lynx suffering from conjunctivitis [1,12,15,16]. Zoonotic transmissions are rarely reported, except for a few cases in which *C. felis* was found to be the cause of keratoconjunctivitis or follicular conjunctivitis in men [19,20,21,22], demonstrating its potential risk for zoonotic infections for humans with close contact to diseased animals in particular [23].

Infected cats usually develop conjunctivitis after an incubation period of 2–7 days [7,24]. The condition often starts as a unilateral process but frequently extends to the other eye and is often characterised by conjunctival chemosis, blepharospasm, ocular discharge, and hyperaemia of the nictitating membrane [4,24,25,26]. Initially, the quality of the discharge is serous but can subsequently become more mucoid to mucopurulent [24]. Some cats may show additional signs such as fever, lethargy, sneezing, serous nasal discharge, as well as submandibular lymph node enlargement, lameness, inappetence, and reduced weight gain (in kittens) [4,25,26,27]. Although some animals recover earlier, most untreated cats develop chronic conjunctivitis with ocular signs usually persisting for 22 to 45 days [7,23,24]. Ocular shedding normally stops after approximately 60 days [24,28]. In some cases, however, Chlamydiae could intermittently be detected up to 8 months post-infection from conjunctival samples of experimentally infected cats, suggesting that part of the cats could remain persistently infected over a longer period representing a population of asymptomatic carriers [4,27,28,29]. Natural transmission requires close contact between infected cats and their aerosols or fomites with ocular secretions being considered the most important infection source [4,24]. Interestingly, Wills et al. (1987) reported that experimental ocular infection of cats led to chlamydial shedding from the vagina and rectum in 50% and 40% of the cases, respectively, demonstrating that *C. felis* is not restricted to the conjunctival mucosa. These authors hypothesised that primary ocular chlamydiosis may result in persistent infection of the genital as well as the gastrointestinal tract [27]. Later, other authors confirmed the detection of *C. felis* in the reproductive tract of experimentally and naturally infected cats, but neither the possibility of venereal transmission nor its epidemiological relevance has been clarified yet [6,7,23,28,29]. These findings support the hypothesis that the faecal–oral route might be an alternative infection pathway for *C. felis*, like it was suggested for different veterinary and human *Chlamydia* species [30,31,32,33]. Despite the above-mentioned evidence for this hypothesis, only a few authors have pursued these findings, and further investigation upon rectal shedding in naturally infected cats is necessary to gain insight into the epidemiology of *C. felis* in the cat population [6,25,34].

In the past, different methods have been used to investigate chlamydial infections in cats. By comparison, PCR of conjunctival swabs showed higher sensitivity than swab isolation and serology [6,34]. Due to the obligate intracellular nature of *Chlamydiaceae*, sufficient numbers of host cells must be collected in the sampling process to ensure detection of the pathogen. In previous studies, cytology brushes were often reported as collection devices for conjunctival samples, but flocked swabs have been successfully used as well [13,35,36,37,38]. Yet, some authors claimed that cytology brushes appear to collect more cells and would therefore be associated with increased detection rates [39,40]. Hence, the conjunctival cell yields of both sampling devices need to be compared.

According to several field studies from different countries (using PCR, isolation, or immunofluorescence assays), the chlamydial prevalence in pet cats ranges from 0% to 10% in healthy animals [34,40,41,42,43,44,45,46] and 5.6% to 30.9% in cats with conjunctivitis [34,40,41,42,43,44,45,47,48,49]. In stray cat populations, the prevalence is usually higher with overall positivity rates of 24.4% to 35.7% [34,46,48,50,51] but can reach up to 65.8% in subgroups with conjunctivitis [48]. One study investigating the occurrence of chlamydial infections in Swiss pet cats in 2003 found that 11.5% of the cats with conjunctivitis and 3.3% of healthy cats had positive conjunctival swabs for *C. felis* [45]. However, no further studies have been carried out to pursue these findings, and data on chlamydial infections in the local stray population is lacking. In order to assess the health status of stray cats in Switzerland and gain information on potential zoonotic hazards, further investigations are needed.

Therefore, the objectives of the present study were to (i) compare the feline cell yields of conjunctival samples obtained with flocked swabs and cytology brushes, (ii) collect data on the occurrence of chlamydial infections in Swiss stray and pet cats, (iii) identify the involved *Chlamydia* species including typing of *C. felis*, and (iv) examine whether *Chlamydiaceae* are shed from the gastrointestinal tract of naturally infected cats.

## 2. Results

### 2.1. Flocked Swabs Led to Higher Feline Cell Yields Compared to Cytology Brushes

We aimed to compare two types of swabs regarding their yields of feline albumin DNA (single-copy reference gene) as a representative value for the total amount of sampled feline DNA and the number of conjunctival epithelial cells based on quantitative PCR (qPCR) [50]. The mean cycle threshold (µ Ct) values of sample duplicates ranged from 22.3 to 29.2 in flocked swabs and 24.8 to 32.1 in cytology brushes (Appendix A). The differences in μ Ct values between paired samples of different swab types were calculated and found to be normally distributed when tested by a normal Q-Q plot (Appendix A) and the Shapiro–Wilk normality test (*p* = 0.446). Ct values of the two groups were significantly different (Figure 1) with a mean difference of 2.5 ± 1.8 (median = 2.5), showing lower Ct values and therefore higher DNA yields in the flocked swabs group. A power analysis confirmed that the number of sampled animals was sufficient to support this result (power = 0.9999995). All results of the statistical analysis are presented in the Appendix A.

### 2.2. Chlamydiaceae 23S rRNA qPCR

#### 2.2.1. *Chlamydiaceae* Were More Often Detected in Symptomatic Cats

The results of the *Chlamydiaceae* family-specific qPCR, which was performed as the initial screening method for conjunctival and rectal samples of 395 cats, are presented in Table 1. Conjunctivitis was diagnosed in 65/86 pet cats (75.6%), whereas 21/86 animals (24.4%) did not show such signs at the time of sampling and were therefore classified as asymptomatic. Furthermore, signs of conjunctivitis were present in 67/309 stray or feral cats (21.7%). While stray cats are socialised to humans, feral cats are not. However, both groups have no or limited contact to humans and were therefore combined to a single group only referred to as stray cats. The ocular health status of three stray cats (1.0%) was unknown, while the remaining 239/309 stray cats (77.3%) were classified asymptomatic. *Chlamydiaceae* were detected in 49/132 symptomatic (37.1%), 18/260 asymptomatic (6.9%), and 2/3 uncategorised cats, resulting in a total of 69 positive cats in 395 sampled animals (17.5%). *Chlamydiaceae* were more often detected in symptomatic cats (Pearson’s Chi-squared test, *p* = 8.953 × 10^−14^). Interestingly, noticeable differences were also observed concerning the occurrence of conjunctivitis within stray cats of different Swiss cantons. Whereas 28.6% of cats in Nidwalden (NW) and 25.0% in Obwalden (OW) showed signs of ocular infection, only 8.8% and 5.9% of cats in Fribourg (FR) and Berne (BE) were symptomatic.

#### 2.2.2. The Prevalence of *Chlamydiaceae* Differs between Symptomatic Stray and Pet Cats but Not between the Overall Stray and Pet Populations

Regarding the entire sample population consisting of symptomatic and asymptomatic animals, *Chlamydiaceae* were detected in 59/309 stray cats (19.1%) and in 10/86 pet cats (11.6%). The positivity rates for stray and pet cats showing signs of ocular infection were 59.7% and 13.8%, respectively. Whereas 7.1% of healthy appearing stray cats were positive, *Chlamydiaceae* could only be detected in one of the asymptomatic pet cats (4.8%). Overall, the chlamydial positivity rates of stray and pet cats did not differ significantly (Pearson’s Chi-squared test, *p* = 0.155); however, a statistically significant difference was observed when comparing only symptomatic cats of both groups with each other (Pearson’s Chi-squared test, *p* = 4.956 × 10^−8^), indicating that the chlamydial occurrence in cats with conjunctivitis was greater among the stray population compared to pet cats.

#### 2.2.3. Rectal Shedding Was Observed in 25.0% of *Chlamydiaceae*-Positive Animals

We aimed to investigate rectal shedding of *Chlamydiaceae* in our study population to gain information about the gastrointestinal tract as a possible reservoir for these bacteria. Results of the *Chlamydiaceae* qPCR for different anatomical sampling sites are presented in Table 2. Out of all collected swab samples, *Chlamydiaceae* were detected in 98/565 conjunctival (17.3%), 16/387 rectal (4.1%), and 2/2 swabs of unknown sampling site. Out of 384 cats with complete sets of samples consisting of a rectal and a conjunctival swab of both eyes (pooled or individual), 64 (16.7%) were *Chlamydiaceae*-positive in at least one swab sample by qPCR screening. In detail, 62 cats (16.1%) were only positive in the ocular sampling site (meaning they had one or two positive conjunctival swabs), and 16 cats (4.2%) had a positive rectal swab. Accordingly, the ocular localization was significantly more often positive compared to the rectal sampling site (Pearson’s Chi-squared test, *p* = 1.805 × 10^−8^). Of the 64 *Chlamydiaceae*-positive cats, 48 animals (75.0%) were only positive in the ocular sampling site, two cats (3.1%) only had positive rectal samples, and in 14 cats (21.9%) *Chlamydiaceae* were detected at both sampling sites. The 23S rRNA qPCR resulted in mean Ct values of 31.2±3.5 and 30.6±4.6 for conjunctival and rectal swab samples, respectively. Chlamydial genome copy numbers per swab ranged from 3.40 × 10^1^ to 5.58 × 10^6^ in conjunctival samples (mean = 1.85 × 10^5^ ± 7.73 × 10^5^) and from 7.32 × 10^1^ to 6.55 × 10^5^ in rectal samples (mean = 1.45 × 10^5^ ± 4.69 × 10^3^) (Figure 2). Results of the chlamydial quantity calculation are presented in the Appendix A.

### 2.3. Chlamydia felis Is the Most Prevalent Chlamydial Species in Stray and Pet Cats

All *Chlamydiaceae*-positive samples (*n* = 116) were subjected to a DNA microarray assay for identification of the involved chlamydial species. Thereby, 86 (74.1%) samples were recognised as *C. felis*, one (0.9%) as *C. abortus* (sample DD2.2e), 15 (12.9%) were negative, and 14 samples (12.1%) were assigned to the *Chlamydia* group containing *C. abortus*, *C. psittaci*, *C. felis*, *C. avium*, *C. gallinacea*, *C. caviae*, *C. pneumoniae*, *C. ibidis*, and *C. pecorum* but could not be further classified. By DNA microarray assay, three of the negative and 12 of the unclassified *Chlamydia* samples showed an inconclusive binding pattern as only one of the two *C. felis* probes (Cp-felis_2) showed a signal above the threshold, whereas the other probe (Cfel_Baker_535) did not. No mixed infections were detected by the DNA microarray assay. The 29 samples that were negative or had inconclusive results were further analyzed with a conventional PCR based on a partial sequence of the 16S rRNA gene and subsequent DNA sequencing. Thereby, 21/29 samples (72.4%) were identified as *C. felis* by BLASTn analysis (100% nucleotide identity (NI) with the *C. felis* type strain Fe/C-56, AP006861.1), and 2/29 samples (6.9%) were identified as *C. abortus* (samples LD1.2e; 98.9% NI with the *C. abortus* strain GN6, CP0211996.1 and LD4.1e; 96.31% NI with *C. abortus* strain 84/2334, CP031646.1), while six samples could not be further classified. In conclusion, 107 out of 116 positive samples (92.2%) were identified as *C. felis*, and in 3/116 samples (2.6%), the species could be confirmed as *C. abortus*. The *Chlamydia* species of 6/116 samples (5.2%) remained unknown since they produced no or low-quality sequences that did not allow further species identification by BLASTn analysis. Overall, *C. felis* and *C. abortus* were identified in 66/69 (95.7%) and 3/69 (4.3%) *Chlamydiaceae*-positive cats, respectively. Excluding five animals with missing swabs, *C. felis* was detected in the ocular sampling site of 59/61 (96.7%) and the rectal sampling site of 10/61 (16.4%) *C. felis*-positive cats with complete sets of samples.

### 2.4. C. felis pmp9 Typing and Sequencing

We aimed to gain insight into the genetic variability of *C. felis* by analyzing a partial sequence of the *pmp9* gene sequence, which encodes for a polymorphic membrane protein, and which has been shown to be a gene with some genetic variability in *C. felis* [14]. The *pmp9* conventional PCR was performed on 64 conjunctival and 10 rectal swab samples (total *n* = 74) of 63 cats (*n* = 53 stray and *n* = 10 pet cats). The samples were chosen according to the criteria that are described in Section 4.7. Amplification products of 71 samples were subjected to Sanger sequencing, which succeeded in 69 cases. Thereof, 68 samples (98.6%) shared 100% NI with the *C. felis* type strain Fe/C-56 (AP006861.1), while one sample (1.4%) showed a nonsynonymous single-nucleotide polymorphism (SNP) leading to an amino acid change from isoleucine (I) to valine (V) compared to the type strain Fe/C-56 in codon 322 (I322V). This sample originated from a stray cat (MM7) that showed signs of conjunctivitis. Although all three samples from this cat were *Chlamydiaceae*-positive, the SNP was only detected in the left conjunctival sample, while the sequence of the right conjunctival sample shared 100% NI with the *C. felis* type strain Fe/C-56 (AP006861.1). The rectal sample was *Chlamydiaceae*-positive but could not be further classified.

## 3. Discussion

### 3.1. Swab Comparison Trial

The feline albumin gene has been described as a useful single-copy reference gene that can be used to quantify feline genomic DNA [50]. We therefore used the feline albumin qPCR assay described by Helfer-Hungerbuehler et al. to quantify feline genome equivalents (used as a proxy for host cell yields) [50]. The use of flocked swabs resulted in higher host cell yields compared to cytology brushes, which is in line with previous studies where flocked swabs were successfully used for detecting ocular chlamydial infections in pigs [35] and humans [38]. We did not investigate whether using either sampling device had an effect on the chlamydial detection rate; therefore, the biological relevance of our finding is unclear. However, we still recommend the use of flocked swabs over cytology brushes, not only due to higher cell yields but also regarding animal welfare and refinement of animal experimentation procedures (3R principles described by Russle & Burch, 1959) [51]. In our experience, flocked swabs were more comfortable as the material is softer compared to cytology brushes, and collecting conjunctival swab samples was easier due to the smaller swab size.

### 3.2. Chlamydiaceae qPCR

#### 3.2.1. Comparing of Symptomatic and Asymptomatic Cats

Chlamydial infections were significantly more often detected in cats with signs of conjunctivitis (37.1%) compared to healthy appearing animals (6.9%), which is in accordance with previous studies [40,41,42,43,46,52]. In our study, we focused on *Chlamydiaceae* as a causative agent of infectious conjunctivitis and did not test for other infectious agents, which might be of interest for future studies.

#### 3.2.2. Pet Cats

*C. felis*-positive samples were detected in 13.8% of pet cats with conjunctivitis and in one asymptomatic pet cat (4.8%). A previous study conducted in Switzerland in 2003 found similar *C. felis* positivity rates in symptomatic (11.5%) and healthy pet cats (3.3%) [45]. Considering the that the differences between the two studies are only small and can probably be attributed to data variation resulting from small sample sizes and the veterinarian assessing the presence or absence of conjunctivitis, it can be assumed that the occurrence of chlamydial infections in Swiss pet cats has not changed drastically over the years.

#### 3.2.3. Stray Cats

We reported that 19.1% of all sampled stray cats from Switzerland were *Chlamydiaceae*-positive, yet the occurrence fluctuated among different Swiss cantons with higher rates in the cantons NW and OW (25.3% and 21.9%) and lower rates in the cantons BE and FR (5.9% and 7.0%). In general, these differences might be attributed to variations in the population density as well as the nutritional and overall health status of stray cats [46,53]. Studies from other countries also showed large variations between different stray cat populations with positivity rates of around 5% in healthy cats and 23.3% to 65.8% in symptomatic animals [48,52]. This is in line with the findings of the present study where 59.7% of symptomatic and 7.1% of asymptomatic Swiss stray cats tested positive for *Chlamydiaceae*.

#### 3.2.4. Comparing Stray and Pet Cats

The positivity rate for chlamydial infections was comparable in the overall stray population (19.1%) and the pet cats (11.6%) but significantly higher in symptomatic stray cats (59.7% compared to 13.8% in symptomatic pet cats). We assume that direct contact between animals and therefore chlamydial transmission occur more frequently in stray cats roaming freely compared to pet cats, which might also be housed indoors. Moreover, symptomatic pet cats could have been pre-treated with topical or systemic antibiotics, which might not have been reported in the clinical history. Testing for other infectious agents was not performed here but could provide further clues in the future as to why chlamydial infection rates were different between symptomatic subgroups in stray and pet cats. Barimani et al. reported similar positivity rates for feral and companion animals of 26.9% and 22.2%, respectively [54]. However, this contrasts with another study, which detected significantly higher chlamydial rates in stray cats (35.7% compared to 6.0% in pet cats) [46], which was suspected to be caused, at least in part, by inadequate nutrition and possibly a weakened immune system in the unowned cats [46]. It is important to note that, for the current study, stray cats were sampled without any preference, whereas pet cats with conjunctivitis were more likely to be included in the study than healthy animals. Therefore, the chlamydial positivity rate in the pet cat population might be an overestimation. On the other hand, similar rates of chlamydial infections in Swiss stray and pet cats could, to some extent, be attributed to the personal impression that the local stray population is well-nourished and generally seems to be in good health (data not shown), which is in contrast to cats in Slovakia [46].

#### 3.2.5. Risk Factors

Since *C. felis* has poor environmental persistence, direct contact is needed for transmission [24,52]. It has been shown in the past that higher positivity rates could be associated with certain environmental risk factors such as outdoor access and living in a multi-cat environment, which increases the number of contacts between cats [22,44,46]. Halánová et al. confirmed that a higher density of animals is most likely associated with higher chlamydial infection rates, and, according to Hwang et al., the prevalence of pathogens that are transmitted through direct contact is higher in areas where stray cats are being fed and therefore gather in groups [46,53]. Usually, stray cats roam freely and often several animals live on the same farm, perhaps being fed in highly frequented areas, which could explain higher rates of chlamydial infections in stray cats compared to pet cats. We did not investigate the possible impact of the sampling season or age distribution on chlamydial positivity rates since most of the cats were sampled in October and November and because the age of stray cats could only be roughly estimated. However, it has been reported elsewhere that chlamydial occurrence in cats is usually higher in the spring and summer months [43] and that chlamydial infections are more common in cats younger than one year old [4,36,48,51].

#### 3.2.6. Rectal Shedding of *Chlamydiaceae*

*Chlamydiaceae* have been detected in faecal samples of birds and different mammals such as cattle, sheep, pigs, mice, and humans [35,55,56,57,58,59], while clinical manifestation or associated tissue lesions in the gastrointestinal tract were often absent [33,56,60,61,62]. It was therefore hypothesised that *Chlamydiaceae* frequently colonise the gastrointestinal tract of their hosts where they can live as commensals over long periods of times, which some authors explain by the mechanism of persistence [31,32,33,63]. Thus, the gastrointestinal (GI) tract might serve as a chlamydial reservoir and probably represents a source for faecal–oral transmission as well as auto- or re-infection by translocation of the bacteria to other sites of the body [31,32]. In view of these studies, we expected to find a considerable number of asymptomatic cats with rectal chlamydial excretion, similar to the findings of a study in Swiss fattening pigs where the mean chlamydial herd prevalence was 93.0% for rectal swabs [35]. Although we were able to demonstrate that rectal shedding can occur in naturally infected cats, *Chlamydiaceae* were more often detected in the ocular sampling site (positive in 16.1% of sampled cats) compared to the rectal localisation (4.2%). Out of 64 cats that were classified as *Chlamydiaceae*-positive by the qPCR screening, 16 (25.0%) had positive rectal samples; however, most of them (14/16) were also positive in the ocular sampling site, whereas only two cats were exclusively positive in the rectal swab. One of these cats was an asymptomatic stray (no medical record available), and the other one was a seven-month-old pet cat with a history of feline infectious peritonitis, uveitis, glaucoma, and corneal ulceration; yet, signs of conjunctivitis were neither reported for the time of sampling nor for any earlier time point. In the current study, we did not perform chlamydial isolation, which would have been necessary to confirm the presence of viable *Chlamydia* in the rectum. However, chlamydial copy numbers of conjunctival and rectal samples were compared, and they were within the same range (Appendix A), which suggests that active replication happened in the GI tract and that not only chlamydial DNA remnants were detected. Taken together, although we found evidence that *Chlamydia* is able to replicate in the feline GI tract, we suspect that rectal excretion most likely occurred secondary to an ocular chlamydial infection and that no persistent colonisation of the GI tract was established. It is therefore reckoned that the feline digestive tract is not a long-term chlamydial reservoir for most cats, but further studies are needed to confirm these assumptions.

### 3.3. Chlamydia Species Identification

As expected, *C. felis* was the most prevalent *Chlamydia* species with detection rates of 94.9% (56/59) and 100% (10/10) in *Chlamydiaceae*-positive stray and pet cats, respectively. Surprisingly, *C. abortus* was found in the conjunctival swabs of 3/59 positive stray cats (5.1%) (DD2 from FR 2018; LD1 and LD4 both from NW 2020), all of which were symptomatic. This finding was unexpected as, to our knowledge, *C. abortus* has only been reported once in a cat, which was in the context of arteriosclerotic lesions [15]. This pathogen is usually associated with late-term abortions or the birth of weak neonates in sheep and goats, but cattle, pigs, and horses can also be affected albeit to a lesser extent (reviewed in [1,64]). According to Chanton-Greutmann et al., approximately 39% of abortions in Swiss sheep and 23% in goats were attributed to *C. abortus* between 1996 and 1998 [65]. The placental tissue and fluids contain large amounts of bacteria, which are shed into the environment upon abortion or lambing [64]. Since the stray cats in this study came from rural areas and most of them lived on farms, it might be possible that small ruminants were in the vicinity and that the detected *C. abortus* were caused by environmental contamination or eating/licking of placental tissue by the cats. All stray cats were sampled during October and November, which is just prior to the main lambing season (December to May) [66]; however, non-seasonal lambing as well as abortions can occur throughout the year [66], explaining why some of the cats might have entered into contact with the pathogen at that time. Furthermore, low temperatures can prolong the survival of *C. abortus* in the environment (usually several days) to weeks or even months [64]. It cannot be concluded from the qPCR results whether the samples contained viable *C. abortus*, but very low copy numbers of *C. abortus* in all three samples (Appendix A) suggest environmental contamination rather than active infection. Again, chlamydial isolation in cell culture would have been necessary to prove this hypothesis. Previously, a case of *C. abortus*-induced follicular keratoconjunctivitis was reported in a dog, but further studies are necessary to clarify whether *C. abortus* is able to replicate in feline conjunctival cells and if it could also be a cause of feline conjunctivitis [67]. Whereas chlamydial species identification was successful in 110/116 swab samples (94.8%), six *Chlamydiaceae*-positive rectal swabs (5.2%) could not be further classified, possibly caused by insufficient sample quality, though potential inhibition of amplification was ruled out by using an internal amplification control [68]. With the exception of these non-classified samples, our findings highlight the benefit of a two-step approach consisting of a *Chlamydiaceae* screening followed by *Chlamydia* species identification in order to enable the detection of unexpected chlamydial species in the examined hosts.

### 3.4. C. felis Typing

The *C. felis* genome consists of a circular ~1.166 kilobase pair (kbp)-chromosome as well as a ~7.6 kbp-plasmid [69,70]. Previous genotyping attempts revealed that *C. felis* shows a high degree of genomic conservation in the 16S rRNA [71] and following major outer membrane protein (*ompA*) genotyping [72,73] as well as the seven housekeeping genes that are used in the multi-locus sequence typing (MLST) scheme [29,74]. In contrast, multiple locus variable-number tandem repeat analysis (MLVA) has revealed more diversity [29]. Tissue tropism, however, may be insufficiently recognised by MLVA as it was unable to detect a specific pattern in a *C. felis* sample from a placental swab compared to conjunctival samples [29]. Moreover, there seems to be a relatively high potential for homoplasy in different patterns; therefore, the use of MLVA in molecular epidemiological investigations is questionable [29]. Considering these drawbacks of classic typing schemes, we chose a different approach to search for genetic variability of *C. felis* by sequencing a partial sequence of the *pmp9* gene, which encodes for the polymorphic membrane protein Pmp9. We selected *C. felis*-positive samples from different sampling sites (conjunctival and rectal), cat populations (stray and pet cats), as well as different clinical manifestations (symptomatic and asymptomatic cats) for *pmp9* typing. We found that 73/74 (98.6%) partial *pmp9* sequences shared 100% nucleotide identity (NI) with each other and with the *C. felis* type strain Fe/C-56 (AP006861.1). Interestingly, Seth-Smith et al. detected increased SNP densities in the genes *pmp13G* and *pmp16G* of *C. abortus*, which is another *Chlamydia* species with low genetic variability including classic typing schemes such as *ompA* genotyping, MLST and MLVA [75,76]. Harley et al. characterised twelve *C. felis pmp* genes and thereby detected a single-nucleotide polymorphism in the sequence of *pmp9* [14]. In our samples, we also detected one nonsynonymous single-nucleotide polymorphism in a conjunctival sample of a stray cat with conjunctivitis (MM7.1e), but the biological relevance of this mutation remains unknown, and isolation would have been necessary for further investigation. We conclude that the *C. felis pmp9* gene is highly conserved in the sampled cat population, confirming the genomic stability of *C. felis*.

## 4. Materials and Methods

### 4.1. Samples

A total of 954 dry swab samples were collected from 395 cats in different regions of Switzerland between October 2017 and January 2021. In detail, the samples consisted of 565 conjunctival, 387 rectal, and two swabs of unknown anatomical localisation from 309 stray or feral cats (referred to as stray cats) and 86 pet cats (Appendix A). The stray animals were sampled during anaesthesia in the context of trap-neuter-return programs that were organised by the animal welfare organisation NetAP (Network for Animal Protection) and carried out in four different Swiss cantons, namely, Berne (BE), Fribourg (FR), Nidwalden (NW), and Obwalden (OW). Samples from pet cats were obtained during veterinary consultations at several small animal practices in the canton of Zurich (ZH) as well as the ophthalmology unit of the Vetsuisse Faculty, University of Zurich, ZH. The pet cats were presented to the veterinarians for different reasons including ocular problems. Sample collection and assessing the presence or absence of conjunctivitis was carried out by a veterinarian who was either a specialised ophthalmologist (at the ophthalmology unit) or a veterinarian not specialised in ophthalmology (at small animal practices and spaying programs). A cat was classified as symptomatic if one or more of the following signs of conjunctivitis were present at the time of sampling: ocular serous, mucous or mucopurulent discharge, blepharospasm, conjunctival hyperaemia, and chemosis. Permission for animal experimentation was obtained from the Cantonal Veterinary Office Zurich (animal experimentation permit No. ZH084/20). Conjunctival samples of 42 apparently healthy stray cats were collected using two different swab types for comparison of feline cell yields. For this purpose, one randomly chosen eye was sampled with a flocked swab (FLOQSwab^®^, Copan Flock Technologies, Brescia, Italy), whereas a cytology brush (Cepillo cervical cell sampler, Deltalab, Barcelona, Spain) was used for the other eye. In the other cats a pooled swab sample (FLOQSwab^®^, Copan Flock Technologies, Brescia, Italy) from both eyes was collected in asymptomatic animals, whereas both eyes were sampled individually if an animal showed signs of conjunctivitis or any other ocular abnormality. All swab samples were stored at −20 °C until further processing. Stray cats were categorised as either young (<6 months) or adult (>6 months) based on body weight and completion of primary dentition. For each stray cat, the age category, sex, clinical signs of conjunctivitis, canton of origin, and date of sampling were recorded. Owners of pet cats were asked to fill in a questionnaire asking for the date of sampling, address, and general data about their cats (age, sex, castration status, number of cats per household, outdoor access, medical history, as well as the presence and quality of nasal and/or ocular discharge and reddening and swelling of the conjunctiva).

### 4.2. DNA Extraction

DNA was extracted using the Maxwell^®^ 16 Buccal Swab LEV DNA Purification Kit #AS1295 (Promega, Madison, WI, USA) following the manufacturer’s protocol. Extracted DNA was eluted in a 50 µL elution buffer. DNA concentrations and 260/280 and 260/230 absorbance ratios were measured by Nanodrop-1000 V3.8.1 (Witec AG, Lucerne, Switzerland), and samples were stored at −20 °C until further processing.

### 4.3. Feline Albumin qPCR for Swab Comparison Trial

For the swab comparison trial, we included swabs of animals without signs of ocular inflammation (which would have influenced the number of sampled cells due to increased numbers of inflammatory cells). A further exclusion criterion was positivity in the *Chlamydiaceae* 23S rRNA qPCR, which applied to seven out of the 42 cats. Altogether, the feline albumin (fALB) qPCR was performed on 70 paired conjunctival swab samples (*n* = 35 flocked swabs and *n* = 35 cytology brushes) using the ABI 7500 Fast Real-time PCR System (Thermo Fisher Scientific, Waltham, MA, USA) following the protocol described by Helfer-Hungerbuehler et al. (2013) [50]. Briefly, for each reaction, 5 μL of DNA template was added to 20 μL of master mix consisting of 12.5 μL 2X TaqMan^TM^ Fast Universal PCR Master Mix (Thermo Fisher Scientific, Waltham, MA, USA) as well as both primers and the probe (Microsynth, Balgach, Switzerland) in a final concentration of 0.5 μM and 0.2 μM, respectively. Information on primers and probes used for this study is listed in Table 3. The amplification was initialised with 95 °C for 20 s followed by 45 cycles of 95 °C for 3 s and 60 °C for 45 s. All samples were tested in duplicates and the mean Ct value was calculated for every duplicate. In each run, three negative controls and a standard curve for quantification were included [50]. The fALB DNA amplification threshold was automatically adjusted by the PCR system.

### 4.4. Chlamydiaceae 23S rRNA qPCR

Initially, all 954 samples were screened with the 23S rRNA-based *Chlamydiaceae* family-specific real-time PCR [77]. The qPCR was carried out with the ABI 7500 Fast Real-time PCR System (Thermo Fisher Scientific, Waltham, MA, USA) including an internal amplification control based on enhanced green fluorescent protein (eGFP) DNA as previously described [68,78,79,80,81]. Cycling conditions were set to 95 °C for 20 s, followed by 45 cycles of 95 °C for 3 s, and 60 °C for 30 s. Samples with DNA concentrations above 150 ng/μL were diluted 10-fold, and all samples and dilutions were tested in duplicates. Three negative controls and quantified *C. abortus* samples (standard curve) were included in each run. The thresholds for chlamydial and eGFP DNA amplification were set to 0.1 and 0.01, respectively. A sample was classified positive, questionable, or negative if the mean chlamydial threshold cycle (Ct) value of both duplicates was below 38, above 38, or undetermined, respectively. Questionable samples and those with inhibited reactions, indicated by a mean eGFP Ct value over 30, were retested in the original concentration as well as in a 10-fold dilution and classified as mentioned above. Repeatedly questionable samples showing Ct values above 38 for the second time were considered positive. The chlamydial quantity of positive samples was calculated with an exponential function *f*(x) = Ae^bx^ where x indicates the Ct value, *f*(x) equals the chlamydial copy number, and e represents the Euler’s number. The Ct values of the quantified *C. abortus* samples were used for calculating the y-intercept A as well as the exponential constant b.


pathogens-10-00951-t003_Table 3Table 3PCR primers and probes, final primer concentration (final conc.), primer sequences, and amplicon size (size) for the different PCRs and gene targets used in this study.PCRGene TargetPrimer and ProbeFinal Conc.Sequence (5′ to 3′)SizeRef.*Chlamydiaceae* 23S rRNA qPCR23S rRNACh23S-FCh23S-RCh23-S-P0.5 μM0.5 μM0.2 μMCTGAAACCAGTAGCTTATAAGCGGTACCTCGCCGTTTAACTTAACTCCFAM-CTCATCATGCAAAAGGCACGCCG-TAMRA111 bp[80]eGFPEGFP-1-FEGFP-10-REGFP-Hex0.4 μM0.4 μM0.2 μMGACCACTACCAGCAGAACACCTTGTACAGCTCGTCCATGCHEX-AGCACCCAGTCCGCCCTGAGCA-BHQ1177 bp[68,81]Feline albumin qPCRfALBfALB-345FfALB-494RfAlb-413P0.5 μM0.5 μM0.2 μMGATGGCTGATTGCTGTGAGACCCAGGAACCTCTGTTCATTFAM-ATCCCGGCTTCGGTCAGCTG-TAMRA150 bp[50]DNA microarray assay23S rRNAU23-F1923R-220.5 μM0.5 μMATTGAMAGGCGAWGAAGGABiotin-GCYTACTAAGATGTTTCAGTTC175 bp[82]eGFPEGFP-11-FEGFP-10-R0.25 μM0.25 μMCAGCCACAACGTCTATATCATGBiotin-CTTGTACAGCTCGTCCATGC277 bp[68,83]16S rRNA PCR16S rRNA16S-IGF16S-IGR0.3 μM0.3 μMGATGAGGCATGCAAGTCGAACGCCAGTGTTGGCGGTCAATCTCTC278 bp[71,84]*C. felis pmp9* typing PCR
*pmp9*
Pmp9-FPmp9-R0.2 μM0.2 μMGCGATTCATGTAGCAGCAAAGTCCAGCTTCCTTGATACCC632 bp[14]


### 4.5. DNA Microarray Assay

All *Chlamydiaceae*-positive samples (*n* = 116) were further analysed with a DNA microarray assay (Chlam Type-23S AS-4 Kit #246500096, Alere Technologies GmbH, Jena, Germany), which was previously established by Borel et al. (2008) and modified to include an internal amplification control [70,82,83]. Sample preparation included a 5′-biotinylation PCR that was performed with the Biometra TRIO thermal cycler (Analytik Jena AG, Jena, Germany). The product was further processed with the Hybridization Kit (Alere Technologies GmbH, Jena, Germany), and binding signals were measured with the ATR-01 array tube reader (Clondiag Chip Technologies, Jena, Germany) [82].

### 4.6. 16S rRNA Pan Chlamydiales PCR

Negative or unclassified samples by the DNA microarray assay were subjected to a short 16S rRNA pan *Chlamydiales* PCR followed by Sanger sequencing. DNA amplification of the 278 bp product was performed with the Biometra TRIO thermal cycler (Analytik Jena AG, Jena, Germany) according to a protocol based on previously established methods [71,84]. In detail, each reaction consisted of a 3 μL DNA template, 25 μL AmpliTaq Gold^TM^ 360 Master Mix (Thermo Fisher Scientific, Waltham, MA, USA), 19 μL molecular grade water, and 1.5 μL of 16S IGF (forward) and 16S IGR (reverse) primers (10 μM, Microsynth, Balgach, Switzerland) (Table 3) adding up to a total volume of 50 μL. Cycling conditions consisted of initialisation at 95 °C for 10 min followed by 40 cycles of 95 °C for 30 s, 58 °C for 30 s, and 72 °C for 60 s as well as a final extension at 72 °C for 7 min. Positive and negative controls were included in each run. PCR products were separated by agarose gel electrophoresis and PCR-positive samples were prepared for Sanger sequencing (see Section 4.8).

### 4.7. C. felis pmp9 PCR

*C. felis* positive conjunctival and rectal samples were further investigated for *pmp9* typing if the mean chlamydial Ct value of the 23S rRNA qPCR was below 38 for rectal or below 36 for conjunctival samples [14]. For animals with two positive conjunctival swabs, the sample with the lower Ct value was analysed. Additional samples were included regardless of the mean Ct value if they showed an abnormal binding pattern in the DNA microarray assay, such as a missing spike at the *C. felis* Baker strain position, but were identified as *C. felis* with the 16S rRNA PCR. The *C. felis pmp9* PCR was performed on 74 samples with the Biometra TRIO thermal cycler (Analytik Jena AG, Jena, Germany) following a modified protocol described by Harley et al. (2007) [14]. In detail, the reaction mixture consisted of a 3 μL DNA template, 25 μL AmpliTaq Gold^TM^ 360 Master Mix (Thermo Fisher Scientific, Waltham, MA, USA), and 1.5 μL of forward and reverse primers (10 μM, Microsynth, Balgach, Switzerland) (Table 3), resulting in a total volume of 50 μL per reaction. A positive control (kindly provided by Antonietta Di Francesco from the Department of Veterinary Medical Science, University of Bologna, Italy) and negative controls were included in each run. DNA amplification was evaluated with agarose gel electrophoresis, and positive samples were subjected to Sanger sequencing.

### 4.8. Sequencing Preparation and Analysis

Sample preparation for sequencing included purification of 16S rRNA (*n* = 29) and *pmp9* (*n* = 71) PCR products using the Gene JET PCR Purification Kit #K0702 (Thermo Fisher Scientific, Waltham, MA, USA) according to the manufacturer’s instructions. Two separate tubes including either reverse or forward primers were prepared with a final concentration of 1.5 ng/100bp DNA per 1 µL in a total volume of 15 µL. Sanger sequencing was performed by Microsynth (Balgach, Switzerland). The obtained sequences were assembled, and consensus sequences were created with the licensed software Geneious Prime version 2021.0.3 (http://www.geneious.com, accessed between January and March 2021) before comparing them to the NCBI database with the alignment search tool BLAST (https://blast.ncbi.nlm.nih.gov/Blast.cgi, accessed between January and March 2021). The 16S rRNA and *C. felis pmp9* sequences were submitted to the NCBI GenBank (https://www.ncbi.nlm.nih.gov/genbank/, accessed on 15 March 2021) under the accession numbers MW741852–MW741871 and MW756136–MW756209, respectively.

### 4.9. Statistical Analysis

Statistical analysis was carried out with the open-source software R Studio (https://rstudio.com/). For comparing feline cell (DNA) yields of two swab types, the differences in fALB Ct values between paired samples were visually and statistically tested for normal distribution by means of a Q-Q plot and a Shapiro–Wilk test, respectively. Ct values between swab types were compared by paired Student’s *t*-test. A power analysis for the swab comparison was performed using the pwr.t.test function in R Studio. Effect sizes were calculated as recommended by Gibbons et al. 1993 [85]. A Pearson’s Chi-squared test was conducted for comparing positivity rates of (1) symptomatic and asymptomatic cats, (2) stray and pet cats, (3) symptomatic stray and symptomatic pet cats, as well as (4) conjunctival and rectal sampling sites. For all analyses, the significance threshold was set to 0.05.

### 4.10. Ethical Statement

Sampling of animals was conducted in strict accordance with the Swiss law of animal welfare and permission for animal experimentation was obtained from the Cantonal Veterinary Office Zurich (animal experimentation permit No. ZH084/20).

## Figures and Tables

**Figure 1 pathogens-10-00951-f001:**
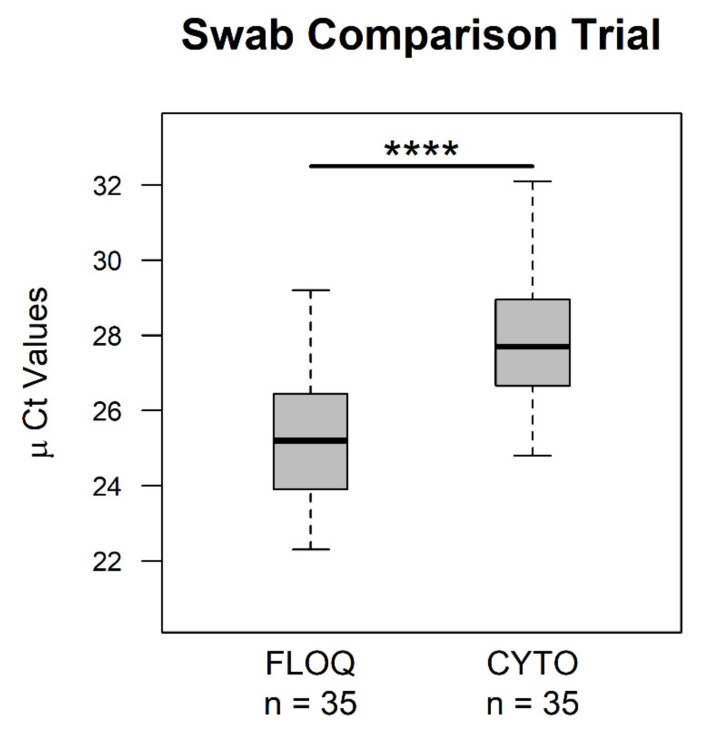
Comparison of Ct values of flocked swabs and cytology brushes resulting from the feline albumin qPCR: Ct values of paired conjunctival swab samples obtained with *n* = 35 flocked swabs (FLOQ) and *n* = 35 cytology brushes (CYTO) differ significantly. Samples were analyzed in duplicates, of which mean Ct values (µ Ct values) are depicted as boxplots. Boxes represent the interquartile ranges from the 25th percentile to the 75th percentile, including the median (bold line), and whiskers show the minimal and maximal values. Statistical analysis was done with paired Student’s *t*-test, **** = *p* < 0.0001.

**Figure 2 pathogens-10-00951-f002:**
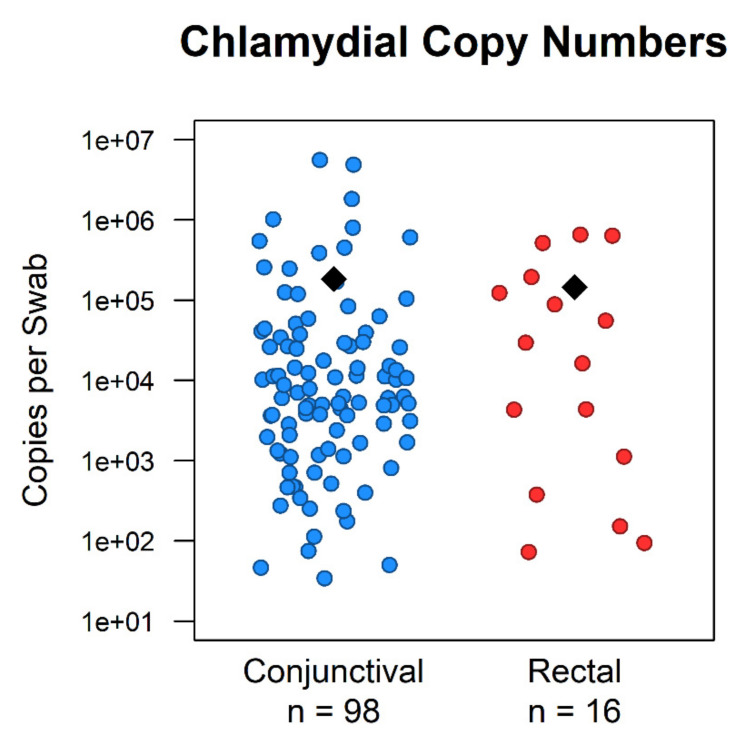
Chlamydial copy numbers of conjunctival (blue dots) and rectal (red dots) swab samples: Chlamydial copy numbers per swab with mean copy numbers (black diamond) of 1.85 × 10^5^ for conjunctival and 1.45 × 10^5^ for rectal samples. The chlamydial copy number was calculated using the same method for samples with known *Chlamydia* species as for *Chlamydiaceae*-positive samples that were not further classifiable.

**Table 1 pathogens-10-00951-t001:** *Chlamydiaceae*-(23S rRNA qPCR) positive/total cats (positivity rates in %) for stray and pet cats sampled in different Swiss cantons.

Population (Year)	Symptomatic Cats		Asymptomatic Cats		Health Status Unknown		Total	
BE (2020)	0/2	(0%)	2/32	(6.3%)	-	-	2/34	(5.9%)
FR (2017/18)	4/5	(80.0%)	0/51	(0%)	0/1	(0%)	4/57	(7.0%)
NW (2017/18/20)	26/44	(59.1%)	11/108	(10.2%)	2/2	(100%)	39/154	(25.3%)
OW (2020)	10/16	(62.5%)	4/48	(8.3%)	-	-	14/64	(21.9%)
**Stray Cats Subtotal**	**40/67**	**(59.7%)**	**17/239**	**(7.1%)**	**2/3**	**(66.7%)**	**59/309**	**(19.1%)**
ZH Ophthalmology (2020/21)	9/53	(17.0%)	1/1	(100%)	-	-	10/54	(18.5%)
ZH Veterinary Practices (2020/21)	0/12	(0%)	0/20	(0%)	-	-	0/32	(0%)
**Pet Cats Subtotal**	**9/65**	**(13.8%)**	**1/21**	**(4.8%)**	**-**	**-**	**10/86**	**(11.6%)**
**Total Cats**	**49/132**	**(37.1%)**	**18/260**	**(6.9%)**	**2/3**	**(66.7%)**	**69/395**	**(17.5%)**

**Table 2 pathogens-10-00951-t002:** *Chlamydiaceae*-(23S rRNA qPCR) positive/total swab samples (positivity rates in %) for different sampling sites in stray and pet cats sampled in different Swiss cantons.

Population (Year)	Conjunctival Swabs		Rectal Swabs		Sampling Site Unknown		Total	
BE (2020)	1/50	(2.0%)	1/32	(3.1%)	-	-	2/82	(2.4%)
FR (2017/18)	5/61	(8.2%)	0/56	(0%)	-	-	5/117	(4.3%)
NW (2017/18/20)	56/209	(26.8%)	8/149	(5.4%)	2/2	(100%)	66/360	(18.3%)
OW (2020)	21/93	(22.6%)	3/64	(4.7%)	-	-	24/157	(15.3%)
**Stray Cats Subtotal**	**83/413**	**(20.1%)**	**12/301**	**(4.0%)**	**2/2**	**(100%)**	**97/716**	**(13.5%)**
ZH Ophthalmology (2020/21)	15/107	(14.0%)	4/54	(7.4%)	-	-	19/161	(11.8%)
ZH Veterinary Practices (2020/21)	0/45	(0%)	0/32	(0%)	-	-	0/77	(0%)
**Pet Cats Subtotal**	**15/152**	**(9.9%)**	**4/86**	**(4.7%)**	**-**	**-**	**19/238**	**(8.0%)**
**Total Cats**	**98/565**	**(17.3%)**	**16/387**	**(4.1%)**	**2/2**	**(100%)**	**116/954**	**(12.2%)**

## Data Availability

All data supporting reported results are contained in the manuscript and the Appendix A and are available on the server of the Institute of Veterinary Pathology.

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
