# Peer review of "Occurrence of Chlamydiaceae and Chlamydia felis pmp9 Typing in Conjunctival and Rectal Samples of Swiss Stray and Pet Cats"

_pathogens, 2021, doi:10.3390/pathogens10080951_

Round 1
Reviewer 1 Report
The authors reported occurrence of chlamydia in stray and pet cats in Switzerland. Feline chlamydial infection is caused by Chlamydia felis. In this manuscript the authors detected Chlamydiaceae as a screening and then identified the detected agents. The authors listed four objectives including comparison of sampling swabs and brushes, the occurrence of chlamydial infection in Swiss stray and pet cats, identification and typing of C. felis and shedding of Chlamydiaceae from the gastrointestinal tract of naturally infected cats. All of these four objectives were scientifically examined. The results presented in the manuscript have enough values for chlamydial research. Minor comments are as follows:
- One of their objectives is typing of C. felis based on pmp9. However, introduction does not contain background information about C. felis pmp9. If the information about pmp9 were included, readers can understand the manuscript better.
- L162: What was ‘complete sets’? Detailed description is needed.
- L205: What does the number 61 mean? The number 61 could not be found in Table1. Please clarify the number 61.
- L211: The authors described “… on 64 conjunctival and 10 rectal swab samples”. When a reader read Materials and Methods lines 503-505, the reader understands the meaning of “64”. A brief description on the samples should be included here.
- L470: The authors described on eGFP DNA amplification. Why was the eGFP DNA used in this assay. Detailed description on the meaning of eGFP should be included.
- L489: “4.6.16. S rRNA” should be “4.6. 16S rRNA”.
Author Response
Reviewer #1 (Comments and Suggestions for Authors):
The authors reported occurrence of chlamydia in stray and pet cats in Switzerland. Feline chlamydial infection is caused by Chlamydia felis. In this manuscript the authors detected Chlamydiaceae as a screening and then identified the detected agents. The authors listed four objectives including comparison of sampling swabs and brushes, the occurrence of chlamydial infection in Swiss stray and pet cats, identification and typing of C. felis and shedding of Chlamydiaceae from the gastrointestinal tract of naturally infected cats. All of these four objectives were scientifically examined. The results presented in the manuscript have enough values for chlamydial research. Minor comments are as follows:
- One of their objectives is typing of C. felis based on pmp9. However, introduction does not contain background information about C. felis pmp9. If the information about pmp9 were included, readers can understand the manuscript better.
- L162: What was ‘complete sets’? Detailed description is needed.
- L205: What does the number 61 mean? The number 61 could not be found in Table1. Please clarify the number 61.
- L211: The authors described “… on 64 conjunctival and 10 rectal swab samples”. When a reader read Materials and Methods lines 503-505, the reader understands the meaning of “64”. A brief description on the samples should be included here.
- L470: The authors described on eGFP DNA amplification. Why was the eGFP DNA used in this assay. Detailed description on the meaning of eGFP should be included.
- L489: “4.6.16. S rRNA” should be “4.6. 16S rRNA”.
Author's Response:
- The text has been modified following the reviewer’s suggestion.
- The expression “complete sets of samples” was used for cats of which we received swabs of both eyes (pooled or individual) and the rectum and that therefore had no missing samples. The text has been modified for better understanding.
- The number 61 refers to the number of C. felis-positive cats that had no missing samples. Table 1 does not distinguish between cats with and without missing samples therefore the number 61 does not appear there. The text has been adapted for more clarity.
- The text has been modified following the reviewer’s suggestion.
- Enhanced green fluorescent protein (eGFP) DNA was used as an internal amplification control for PCR. This information has been added to the text.
- The text has been modified following the reviewer’s suggestion.
Reviewer 2 Report
Please see attached.

Author Response
Reviewer #2 (Comments and Suggestions for Authors):
Title: Occurrence of Chlamydiaceae and Chlamydia felis pmp9 typing in conjunctival and rectal samples of swiss stray and pet cats
The general study design was to utilize PCR screening on conjunctival and rectal samples from stray and pet cats, followed by species ID and typing using the highly conserved pmp9 gene. The authors provide a comprehensive review of clinical signs, transmission and shedding of Chlamydia in cats. They went on to describe 4 objectives of this study: sensitivity of sample type (flocked swabs vs cytology brushes), prevalence of chlamydia infections in stray and pet cats, identify the Chlamydia species causing infections (including typing of C. felis), and evaluate rectal samples in relation to conjunctival swabs.
The authors adequately address their objectives, and make logical conclusions based on their data. They used multiple approaches to answer their questions and provided reasonable justifications. The
manuscript is easy to follow and I have no major concerns or suggestions. It was a pleasure to read a
manuscript that the authors put a lot of effort into.
A few specific edits are below:
129-130: edit “….qPCR which was performed as the initial screening method….”
133 and 407: what is the difference between stray and feral cats?
236: “better fitted” is awkward wording
286: italicize C. felis
474-475: samples with Ct values above 38 are positive? In lines 470-472, it is stated below 38 is
considered positive.
489: Should be “16S rRNA pan Chlamydiales PCR”.
Author's Response:
Stray and feral cats both live outdoors and have no or limited contact to humans which is an important factor that distinguishes them from pet cats however, stray cats are socialised to humans while feral cats are not. This information has been added to the text.
In the first run, a sample is considered positive if the Ct value is below 38 and questionable if it is above 38. Questionable samples must be retested and are categorised as negative if the Ct value is undetermined in the second run however, in any other case with the Ct value below or above 38 it is considered positive.
The text has been modified following the reviewer’s suggestion.